# Single Residue Substitution at *N*-Terminal Affects Temperature Stability and Activity of L2 Lipase

**DOI:** 10.3390/molecules25153433

**Published:** 2020-07-28

**Authors:** Noramirah Bukhari, Adam Thean Chor Leow, Raja Noor Zaliha Raja Abd Rahman, Fairolniza Mohd Shariff

**Affiliations:** 1Enzyme and Microbial Technology Research Centre, Faculty of Biotechnology and Biomolecular Sciences, Universiti Putra Malaysia, Serdang 43400, Selangor, Malaysia; noramirahbukhari@gmail.com (N.B.); adamleow@upm.edu.my (A.T.C.L.); rnzaliha@upm.edu.my (R.N.Z.R.A.R.); 2Department of Cell and Molecular Biology, Faculty of Biotechnology and Biomolecular Sciences, Universiti Putra Malaysia, Serdang 43400, Selangor, Malaysia; 3Department of Microbiology, Faculty of Biotechnology and Biomolecular Sciences, Universiti Putra Malaysia, Serdang 43400, Selangor, Malaysia

**Keywords:** lipase, thermostability, rational design, stability prediction, homology modelling

## Abstract

Rational design is widely employed in protein engineering to tailor wild-type enzymes for industrial applications. The typical target region for mutation is a functional region like the catalytic site to improve stability and activity. However, few have explored the role of other regions which, in principle, have no evident functionality such as the *N*-terminal region. In this study, stability prediction software was used to identify the critical point in the non-functional *N*-terminal region of L2 lipase and the effects of the substitution towards temperature stability and activity were determined. The results showed 3 mutant lipases: A8V, A8P and A8E with 29% better thermostability, 4 h increase in half-life and 6.6 °C higher thermal denaturation point, respectively. A8V showed 1.6-fold enhancement in activity compared to wild-type. To conclude, the improvement in temperature stability upon substitution showed that the *N*-terminal region plays a role in temperature stability and activity of L2 lipase.

## 1. Introduction

Lipases have diverse applications in industrial processes that demand enzymes with high stability and tolerance towards extreme conditions. Thermostability and catalytic efficiency are favourable characteristics of enzymes, as many industrial processes occur at high temperatures to facilitate the processes and reduce the risk of biological contamination [1]. The stability and robustness of thermostable lipases make the enzyme appealing to industries such as leather tanning and in the processes of removing pitch in the pulp and paper industry [2,3,4]. Lipases are also known to catalyse reactions such as hydrolysis and esterification, making them valuable in the oleochemical industry and fats and oil modifications [5,6]. Therefore, significant interest exists in using thermostable lipases as these enzymes are more resistant towards high temperatures in addition to a broad catalytic selectivity and capacity.

The thermostability of enzymes is primarily contributed by molecular interactions formed between atoms of residues within the enzyme structure [7,8,9]. Commonly reported of such interactions are hydrogen bonds, ion pairs and hydrophobic bonds [10,11,12]. The increase in such interactions contributes to the compactness of the structure, thus increasing stability at high temperature [3,5,13,14,15]. Other interactions, such as aromatic interaction and disulphide bridge were also discussed as contributing factors in protein stability [7,8,9]. These interactions are aimed by studies to improve stability and in some, the activity of enzyme, through protein engineering strategies such as directed evolution and rational design to satisfy the increasing need for producing more stable lipases [16,17,18,19]. Each strategy has its advantages; however, the rational design approach has been of choice due to its simplicity with the use of computational tools and site-directed mutagenesis [20].

Typical target regions for improvement of lipase stability are the ones directly involved in the catalytic function and stability. However, aiming for interactions in the catalytic site and modification at the lid region requires many considerations [18,19]. Moreover, in the studies of improving stability by introducing disulphide bridge, the location of the cysteine residues must satisfy the geometric constraints for disulphide bonds to form and avoidance from bridge formation with native cysteine [16,17,21]. Whereas the terminal region with no apparent functional role poses less risk but showed possible influence as shown by some studies [11,14,22].

Previously, L2 lipase from thermophilic *Bacillus* sp. L2 isolated from a hot spring in Slim River, Perak was discovered able to remain active between 55 to 80 °C with pH stability range from pH 6.0 to 10.0 [23]. The structure of L2 lipase was solved using X-ray crystallography and showed a globular α/β hydrolase fold [24]. L2 lipase is part of the I.5 lipase family, characterised by its unique helix lid covering the catalytic pocket. The lid manages the access of the substrate into the catalytic site by changing its conformation. This change is activated when the lipase encounters a lipid–water interface in an event called interfacial activation. The lid becomes displaced into an open conformation when in contact with the water–lipid interface, thus allowing the entrance of the substrate into the catalytic site [4,25,26]. Other features include the Zn^2+^ binding motif, *α/β* hydrolase fold and capability of hydrolysis at temperatures 60 to 75 °C in alkaline conditions [27,28,29]. The terminal tails of enzymes can have functional roles such as catalytic domains, transmembrane domain, and intramolecular chaperon [30,31]. In L2 lipase, the terminal tails do not have a defined role, similar to P1, L1, BTL2 and T1 of the I.5 lipase family [28,29,32,33].

Through stability prediction, a previous study by Sani et al. [34] explored the relationship between the lipase structural features and thermostability by substitution of a single residue at a critical point in the C-terminal region of wild type (wt) L2 lipase. It was shown that the substitution increased the optimum temperature of L2 lipase by 10 °C, elongated the half-life at 60 °C and increased the thermal denaturation point by 19 °C. In this study, the role of the *N*-terminal region of L2 lipase towards temperature stability and activity was explored. Rational design using stability prediction tools were employed to determine the critical point. Site-directed mutagenesis was done for the construction of mutants, followed by expression, purification and characterisation.

## 2. Results

### 2.1. Screening for N-terminal Mutation of wt-L2 Lipase

In this study, the *N*-terminal region of L2 lipase was defined as amino acid residues from Ala1 to Gly20 and screening was done to identify the critical point for substitution. The first round of screening with I-Mutant2.0 showed 11 potentials for substitution that exhibited an increase in stability change upon amino acid residue substitution. The molecular interactions at these 11 points were analysed with PIC web server to identify the point with most interaction, of which Ala8 showed the most with 6 total interactions (red arrow in Table 1). The amino acid substituents for Ala8 was determined with a second round of screening with I-Mutant2.0 (Table 2). Nineteen amino acid residues were substituted at Ala8. The substitution showing increase stability change and Reliability Index (RI) score of more than 2 were selected for in silico analysis and experimental validation. The residues for substitutions were valine (A8V), proline (A8P) and glutamic acid (A8E), as indicated by the red arrows in Table 2.

### 2.2. Comparison of Modelled Structures of wt-L2 and Mutant Lipases

The structure of mutants A8V, A8P and A8E were predicted through homology modelling using the remodelled structure of wt-L2 lipase as template. The modelled structures of wt-L2 with A8V, A8P and A8E were superposed for comparison. The results showed that the mutant lipases have similar RMSD values, which were 2.313 Å, 2.354 Å and 2.323 Å respectively over 5,973 matched atoms. The superposition showed several differences throughout the modelled structures, but the most notable difference is the anchoring of the *N*-terminal tail in which there was no overlapped of the tail observed between wt-L2 and its mutants. Further analysis revealed changes in the molecular interactions of critical point 8 and the surrounding amino acid residues that resulted in the folding of the tail towards the global structure compared to that in wt-L2 (Figure 1). The folding or anchoring of terminal tails is typically achieved through ion pairing, hydrogen bonds or hydrophobic interactions [35]. These interactions anchor the flexible terminal tail, consequently reducing the probabilities of the overall structure from unfolding since it typically begins at the terminal tails of protein [35,36]. Further analysis of the *N*-terminal tail of the mutant lipases reveals hydrogen bonds and hydrophobic interactions between the residues of *N*-terminal with the *C*-terminal residues and surrounding residues that are responsible for the anchoring (Table 3). A8E showed the most interactions followed by A8P then A8V.

Superposition of the active site of the modelled mutant lipases reveals no significant change to the position of the catalytic site residues compared to wt-L2. Further study was done to examine the changes in the molecular interactions involving the catalytic residues Ser113, His358 and Asp317. Based on Figure 2, wt-L2 and A8E have 2 hydrogen bonds and 2 hydrophobic interactions. Comparing the distance of hydrogen bond between atom OG Ser113 and atom NE2 His358, the bond in A8E (2.33 Å) is longer than in wt-L2 (2.06 Å) indicating an increase in distance between the catalytic residues. On the contrary, A8V and A8P both have only one hydrogen bond and 2 hydrophobic interactions in the catalytic triad, lacking the hydrogen bond between Ser113 and His358. A study by Lu et al. [36] showed that an increased distance between catalytic residues and loss of hydrogen bonds reduces the rigidity surrounding the catalytic site, thus allowing better access for substrate binding, consequently improving the catalytic activity of the protein. The change in molecular interaction in the catalytic triad may be due to the cumulative effect from the substitution. The cavity volume enclosed by the catalytic triad residues of the modelled lipases were investigated and showed that the volume of wt-L2 and the mutants were similar at 275.93 Å^3^, 275.76 Å^3^, 276.22 Å^3^ and 276.63 Å^3^, respectively. Therefore, there is no significant change to the cavity.

### 2.3. Molecular Dynamics Simulation Analysis for RMSD, RMSF, SASA and Rgyration

The RMSD and RMSF from the 100 ns simulation were used to analyse the stability of the lipases at 70 °C (Figure 3). A8E scored the lowest average RMSD at 2.27 Å followed by A8P, wt-L2 and A8V at 2.62 Å, 2.65 Å and 2.82 Å, respectively. All lipases were observed with equilibration peaks from 0 to 20 ns with A8E showing the least deviation followed by A8P, wt-L2 and A8V. From 60 to 80 ns, A8V exhibited high peaks, whereas the other lipases remained stable. RMSF analysis showed the flexibility of amino acid residues Val193 to Arg230 corresponds to the high RMSD peaks of A8V. The residues Val193 to Glu219 forms a coil and turn structure connecting two alpha helices whereas the residues Ser220 to Arg230 forms one of the alpha helices.

The lid structure in I.5 family lipase consists of two helices covering the catalytic pocket, namely the α6 and α7 that are connected by a hinge made of a coil and turn structure [29,33]. In the present study, the residues Ser220 to Arg230 corresponds to the α7-helix and residues Val193 to Glu219 corresponds to the hinge that connects the α7-helix with the α6-helix (Figure 4). During interfacial activation, the lids of the lipase become displaced into an open conformation when in contact with the water–lipid interface, thus allowing the entrance of the substrate into the catalytic site [4,25,26]. In the study of opened conformation BTL2, the displacement of α7-helix was enabled by the hinge whereas the α6-helix was from lateral displacement and structural reorganisation [33]. In L1 lipase, the displacement of α6-helix was due to the flexible loop at the *C*-terminal end of the lid helix [28]. The region in the catalytic lid is involved in the conformational change of the lid helix during interfacial activation in which the substrate enters the catalytic site through the opened lid [26]. Therefore, flexibility in this area may affect the catalytic activity of A8V. The *N*-terminal amino acids showed the greatest flexibility amongst all the residues in wt-L2 and mutant lipases. However, compared to wt-L2, the mutant lipases exhibited lower flexibility at the *N*-terminal tail.

Another distinct deviation in RMSD at 27 ns, then from 63 ns to 76 ns was observed in α3-helix. The RMSF analysis showed flexibility corresponding to residues Ala80 to His87, highest in A8V at 5.5 Å followed by A8P, A8E and wt-L2. According to Rahman et al. [24], residues His81 and His87 are part of the Zn^2+^ ion binding domain in wt-L2 (Figure 5). The role of Zn^2+^ ion towards structure stability in thermostable L1 lipase investigated by Choi et al. [37] showed Zn^2+^ ion contributed to the structural stability in maintaining active conformation at high temperature. Other I.5 lipases have also shown the role of Zn^2+^ binding domain in stability and opening of the lid [28,33]. Therefore, the role of Zn^2+^ ion as a structure stabiliser in the mutant lipases may also be affected similarly to the affected α3-helix.

An increase in the SASA values indicates an increase in exposure towards the solvent as a result of protein conformation change through partial unfolding. The SASA analysis of the lipases followed a similar trend to the RMSD analysis whereby A8E showed the lowest average value of 15,400.2 Å, followed by A8P (15,536.2 Å), wt-L2 (15,624.7 Å) and A8V (16,329.3 Å) (Appendix A). The radius of gyration (Rgyration) represents the compactness of the global protein structure over the 100 ns simulation time. Average Rgyration value of A8E was the lowest at 20.36 Å followed by A8P (20.37 Å), wt-L2 (20.38 Å) and A8V (20.56 Å) (Appendix A). A trend was observed in which the compactness of the global structure coincides with the SASA values whereby when the SASA increase, the Rgyration would follow. A compact structure is desirable in a structure to be able to remain stable at high temperatures [38]. However, a too compact structure may impede the lipase ability to perform catalysis [39].

### 2.4. Temperature Stability Characterisation of Mutant Lipases

The mutant lipases showed a similar optimal temperature for activity to wt-L2 at 70 °C (Figure 6a). In the thermostability profile (Figure 6b), at 60 °C, the residual activity of wt-L2 and A8V were highest at 94.60% and 91.87%, respectively. However, at 70 °C, only A8V maintained high residual activity (84.10%), whereas A8P, A8E and wt-L2 were reduced to 59.62%, 54.88%, and 55.10%, respectively. Contrarily, in the half-life study at 60 °C (Figure 6c), A8P has the most prolonged half-life at 12 h and both A8V and A8E at 10 h. The shortest was wt-L2 at 8 h. Interestingly, in the first 2 h of incubation, the residual activity of A8P dropped the lowest then gradually recovering before declining at hour 8, suggesting an equilibration period or thermal activation. Meanwhile, the other lipases started at high residual activity, followed by a gradual decline until hour 16.

The denaturation point (T_m_) analysis of the lipases (Appendix A) showed A8E to possess the highest T_m_ at 73.59 °C, 7.17 °C higher than the reported wt-L2 at 66.73 °C [34]. A8V and A8P had only a small difference of T_m_ at 70.68 °C and 70.19 °C, respectively. Interestingly, in measuring the activity at different temperature treatments, both A8V and A8P had better thermostability and longer half-life respectively over A8E. It is likely, despite being the most stable, catalytic efficiency of A8E was inferior compared to A8V and A8P. The T_m_ of A8E aligns with the Molecular Dynamics (MD) simulation analysis, A8E was the most stable, showing the least deviation and fluctuation in addition to the lowest SASA and Rgyration. In the in silico analysis, A8E showed to have the most hydrogen and hydrophobic interactions anchoring the terminal tail. Terminal tails are flexible sites that have been reported to be the unfolding start site in protein [35]. Therefore, A8E was able to resist unfolding at a higher temperature compared to the other lipases.

### 2.5. Substrate Specificity and Kinetic Constants of wt-L2 and Mutant Lipases

Based on Figure 7, wt-L2 and the mutant lipases showed a similar preference for *p*NP C10. A8V showed the highest activity at 393.08 U/mL, followed by wt-L2 at 280.35 U/mL, A8P at 201.21 U/mL and lowest was A8E at 129.81 U/mL. Additionally, A8V was found to have notable activity for *p*NP C8 and *p*NP C12, whereas wt-L2 and A8P showed considerable activity in *p*NP C8. A8E showed exclusively high activity at *p*NP C10. Substrate *p*NP C10 was employed to study the reaction rate of the lipases at an incubation temperature of 70 °C from times 0 to 10 min with 2 min interval.

Based on Table 4, A8V has the highest catalytic efficiency at 260.57 s^−1^/mM, followed by wt-L2 at 162.43 s^−1^/mM, A8P at 94.93 s^−1^/mM and A8E at 27.23 s^−1^/mM. Although wt-L2 showed higher *kcat* than A8V, the K_M_ of A8V was significantly lower, therefore, contributing to the catalytic efficiency of the mutant lipase. The K_M_ of A8E was drastically the highest compared to other lipases, indicating a low affinity towards the substrate. This may be due to the increase in rigidity of A8E, including the lid region that obstructs the entrance of substrate towards the catalytic site. This resulted in the need for a high concentration of substrate to saturate the catalytic site and coupled with the low *kcat*, reduced the catalytic efficiency. Therefore, based on A8V, the affinity towards substrate takes precedence over the turnover rate. The flexibility in the lid region of A8V according to the RMSF analysis, thereby is speculated to enhance the substrate affinity. 

### 2.6. Optimum pH and pH Stability of Mutant Lipases

Based on Figure 8, the highest activity in A8V and A8P was in Tris-HCl pH 8.0, whereas A8E was highest in sodium phosphate pH 7.0. Shariff et al. [23] reported that wt-L2 showed activity range at pH 6.0 to 10.0 and highest at glycine-NaOH pH 9.0. The mutant lipases, therefore, showed an acidic shift in the catalytic pH compared to wt-L2. 

The stability of the mutant lipases in multiple buffer systems of pH ranging from pH 4.0 to 10.0 was also tested by assaying the residual activity after incubation for 30 min at 60 °C. The pH stability profile in Figure 9 showed all mutant lipases were able to retain activity above 50% from pH 6.0 to 10.0, a broader range compared to wt-L2 lipase, which ranged from pH 8.0 to 10.0 [23]. 

In the catalytic site, the pKa of ionisable catalytic residues plays a vital role in the pH characteristics and catalytic mechanism [40]. The pKa values of these residues are highly dependent on the surrounding environment, such as solvent, electrostatic interactions and hydrogen bonds [41]. Further look at the pKa of the ionizable Asp317 and His358 of the catalytic triad using webserver DEPTH (http://cospi.iiserpune.ac.in/depth/htdocs/index.html) showed pKa values (Appendix A) did not align with the catalytic triad analysis in Section 2.2 and did not explain the shift in the pH towards acidic limb of the pH profile. Therefore, it is postulated that the effects of the mutations towards pH were not from the catalytic site but rather from the overall change in pKa and consequently, the pI and pH. The theoretical pI of the mutant lipases determined using webserver ExPASy (http://web.expasy.org/compute_pi) showed that A8V and A8P have the same pI as wt-L2 at 6.12. Thus, the minor shift of one catalytic pH value is reasonable. A8E, however, has lower pI at 6.01, likely explaining the catalytic pH shifting towards the acidic limb.

## 3. Discussion

The stability prediction for single residue substitution at the *N*-terminal was shown to have an overall improvement to the thermal stability and catalytic activity of L2 lipase. Mutant A8V exhibited enhancement in thermostability at over 80% of the residual activity compared to 50% of wt-L2 at 70 °C. Based on the cavity analysis, there was only a marginal difference in the volume of the catalytic triad. The preference of *p*NP C10 in the lipases may likely be due to the stearic effect of the triad residues. Therefore, it is reasonable the preference of the lipases does not change. Although in A8V, *p*NP C10 was the primary preference, the mutant was also amendable to allow catalysis of varying carbon chain lengths. Moreover, the catalytic efficiency improved 1.6-fold in efficiency compared to wt-L2, 2.7-fold than A8P and 9.5-fold than A8E. 

These changes in the catalytic activity of A8V was most probably due to the flexibility of structures surrounding the lid region and the affected Zn^2+^ domain, as evident in the increase of fluctuations of amino acid residues from the region. Structures in the vicinity of the lid region of other I.5 family lipases were shown to be involved in the catalytic mechanism through molecular interactions from the amino acid residues. In L1 lipase, α6- to α9-helix form a rigid structure on the catalytic site of the lipase, and the opening of α6-helix lid is triggered by thermal dissociation of the hydrophobic interactions [28]. Similarly, in P1 lipase, the α11- and α12-helix bound the lid region and opening of α6-helix lid involved change in positions of hydrophobic residues [32]. In BTL2 lipase, a network of salt bridge interactions was observed in the closed states that directly connect α6- and α7-helix lids that were involved in movements of the lids during opening. The increase in flexibility in the region around the lid, particularly contributed by the highly disordered coil and turn structures, would ease the entrance of the substrate into the catalytic site, consequently enhancing the catalytic activity [36,39]. The structures surrounding the lid of A8V showing an increase in flexibility may ease the opening of the lid, therefore, could collectively contribute to better catalytic activity. 

Interestingly, despite having the better thermostability and catalytic efficiency, the half-life of A8V was shorter than A8P by 2 h and the thermal denaturation point was about 3 °C less than A8E. The thermal stability exhibited by A8P may be contributed by the proline substitution at the non-catalytic *N*-terminal region, resulting in an improvement of lipase stability, thus allowing the mutant lipase to remain active at high temperature for a prolonged time. A study by Mohammadi et al. [31] reported substitution of proline residue at position 2 in the *N*-terminal of lipase A from *Serratia marcescens* revealed a 2.3-fold higher activity at half-life of the mutant compared to the wild-type. Liu et al. [42] also reported an increase of 7-fold in half-life at 45 °C of mutant alkaline lipase I from *Penicillium cyclopium* with proline substitution at position 41, far from the catalytic site. In both studies, the introduction of proline residue improved the stability of the structure, thus prolonging the half-life of the protein.

Although having the highest thermal denaturation point, A8E had the lowest catalytic efficiency. The increase in stability achieved by A8E was at the cost of its activity. Based on MD simulation, A8E was the most rigid enabling it to withstand high temperature but the rigidity may have hindered the movement of the lid for interfacial activation during catalysis. Thus, maintaining a certain degree of flexibility is important. The reduction in catalytic efficiency of A8E in relation to its increase in stability implies a stability–activity trade-off. According to Miller [43], such occurrence is infrequent, particularly in an engineered enzyme. The author also reiterates that there is no encompassing factor in determining the stability–activity trade-off; rather, the trade-off is unique to each enzyme. Regardless, based on the comparison of the lipases in the present study, the trend of increasing stability resulting in the drop of catalytic efficiency is noteworthy. To withstand high temperatures, a rigid structure of the lipases is required to remain stable. However, for lipases to perform their catalytic function, the structure needed a certain degree of flexibility, particularly at the lid region, to accommodate the entrance of the substrate and release of the product.

## 4. Materials and Methods

### 4.1. Rational Design for N-Terminal Mutant Lipases

Amino acid residues from positions 1 to 20 in the *N*-terminal region of L2 lipase were screened for critical points for residue substitution using I-Mutant2.0 (folding.biofold.org/i-mutant/i-mutant2.0.html) [44]. The screening was done at the optimal conditions of wt-L2 lipase [23]. We found 11 positions with potential as the critical point for the residue substitution based on the sign of protein stability free energy change (ΔΔG) and Reliability Index (RI). These 11 positions were further screened using Protein Interaction Calculator (PIC) (pic.mbu.iisc.ernet.in) [45] based on their respective molecular interactions. Position 8, with wild-type residue Ala was chosen as the critical point for substitution with residues Val, Pro and Glu. The amino acid residues chosen for substitution were based on second-round screening with I-Mutant2.0.

### 4.2. Remodeling of wt-L2, Homology Modelling and Molecular Dynamics Simulation of Mutant Lipases

The crystal structure of wt-L2 (PDB ID: 4FDM) deposited in the PDB database is truncated; therefore, it requires remodelling to obtain the complete structure. T1 lipase (PDB ID: 2DSN) with 98% sequence similarity with wt-L2 was used as the template. The remodelled structure of wt-L2 was then used as the template for the homology modelling of mutant lipases, namely A8V, A8P and A8E. The residue substitution of valine, proline and glutamic acid was introduced respectively in the FASTA file at position 8 for the modelling. The software Yet Another Scientific Artificial Reality Application (YASARA, ver. 17.14.17) (CMBI, Nijmegen, Netherlands) was employed for both remodelling and homology modelling of the lipases. All structures were validated using the Ramachandran Plot (https://servicesn.mbi.ucla.edu/SAVES/) and Z-score with SWISS-MODEL QMEAN (https://swissmodel.expasy.org/qmean/). 

Molecular Dynamics (MD) simulation at 70 °C and pH 9.0 was done on the remodelled wt-L2 structure and the modelled structure of the mutants. The simulation was run with parameters set to the command file md_run.mcr and by using AMBER14 force field in simulation cell of dimensions 50 Å × 50 Å × 50 Å with steepest descent energy minimisation within 100 ns of production time. The root mean square deviation (RMSD) values were computed to check for the stability of trajectories for the protein backbone and residues. Additionally, the RMSD analysis also yields the Surface Accessible Solvent (SASA) and radius of gyration (Rgyration) for surface area exposed to solvent and the compactness of global protein structure, respectively. The flexibility of the 100 ns trajectories was studied by computing the root mean square fluctuation (RMSF) value per residue.

### 4.3. Construction of Mutant Lipases

The construction of the mutant lipases was done via site-directed mutagenesis using QuikChange Lightning Mutagenesis kit (Agilent, Santa Clara [SCL], USA) with recombinant wt-L2 in plasmid pET32(B)+ as the template. The PCR amplification was done at 1 cycle of 2 min at 95 °C, 18 cycles of 20 s at 95 °C, 10 s at 60 °C and 3 min 55 s at 68 °C and 1 cycle of 5 min at 68 °C. The plasmid containing the mutant lipases was transformed into expression host *E. coli* BL21(DE3), grown in Luria Bertani (LB) broth for 1 h at 37 °C. The culture was then spread onto LB agar plates supplemented with 100 µg/mL ampicillin and incubated for 16 h at 37 °C. Verification for mutant lipase insert was done with DNA sequencing.

### 4.4. Expression and Purification of wt-L2 and Mutant Lipases

The expression of wt-L2 and mutant lipases were done in 200 mL LB broth supplemented with 100 µg/mL ampicillin at 37 °C with shaking. Induction with 1 mM isopropyl β-_D_-1-thiogalactopyranoside (IPTG) was done when OD_600nm_ reached 0.5, then the incubation was continued for 16 h at 37 °C with shaking. The cells were harvested by centrifugation at 15,000× g for 10 min at 4 °C then dissolved in binding buffer (20 mM sodium phosphate pH 7.4, 500 mM NaCl and 50 mM imidazole) to prepare for cell lysis. Sonication was done at 30 s interval for 6 min with 30% duty cycle output to lyse the cells, followed by centrifugation at 15,000× g for 10 min at 4 °C to separate the crude lysate from the cell debris. The crude lysate was then loaded into the purification column with Ni Sepharose 6 Fast Flow resin (GE Healthcare, Danderyd, Sweden). The unbound protein was washed with the binding buffer followed by the elution of target lipase with elution buffer (20 mM sodium phosphate pH 7.4, 500 mM NaCl and 300 mM imidazole). Eluted fractions were collected then dialyzed with 5 mM sodium phosphate pH 8.0. The purity of the lipases was analysed with SDS PAGE (Appendix A) and the concentration was estimated with Bradford Assay [46].

### 4.5. Characterisation of wt-L2 and Mutant Lipases

#### 4.5.1. Effects of Temperature on wt-L2 and Mutant Lipases

The activity assay of the purified wt-L2 and mutant lipases were measured with olive oil (Bertolli, Italy) emulsified with 50 mM Tris-HCl pH 8.0 as substrate according to a modified Kwon and Rhee assay [47]. The concentration of all lipases was fixed at 1.0 mg/mL for all characterisation.

Optimum temperature: The assay incubation was done at temperatures ranging from 55 to 85 °C with 5 °C intervals for 30 min.

Thermostability: The pre-incubation temperature was done at temperatures ranging from 50 to 80 °C with 5 °C intervals for 30 min. The residual activity of the lipases was assayed at their respective optimum temperature for 30 min. The activity of lipases without pre-incubation treatment was defined as 100%. 

Half-life: The lipases were treated by pre-incubation at 60 °C with residual activity assayed at every 2 h interval. The activity of lipases without pre-incubation was defined as 100%. 

Thermal denaturation point: Circular dichroism (CD) was done using JASCO J-810 Spectrapolarimeter (JASCO, Hachioji, Japan) to determine the thermal denaturation point (T_m_). The measurement was executed in a 10 mm path-length cuvette at wavelength 222 nm at temperature variable ranging from 20 to 100 °C with 1 °C/min heating rate. 

#### 4.5.2. Substrate Specificity of wt-L2 and Mutant Lipases

Lipolytic activity of the lipases on different carbon lengths was determined using modified *para*-nitrophenyl (*p*NP) palmitate assay [48] with 5 mM of substrates *p*NP acetate (C2), *p*NP butyrate (C4), *p*NP octanoate (*C*8), *p*NP decanoate (C10), *p*NP laurate (C12) and *p*NP palmitate (C16). 

#### 4.5.3. Kinetic Constants of wt-L2 and Mutant Lipases

The kinetic constants K*_M_*, Vmax, *kcat* and K*_M_*/*kcat* were determined with substrate *p*NP C10 of concentration 0.6 mM, 1.2 mM, 2.5 mM and 5.0 mM. The assay incubation was done at the respective optimum temperature of the lipases for a time ranging from 2 to 10 min at 2 min interval. The concentration of the product was plotted as a function of time to determine the reaction rate of the Michaelis–Menten plot. The inverse of the Michaelis–Menten plot was then used to construct the Lineweaver–Burk plot of which the constants mentioned above were derived. 

#### 4.5.4. Optimum pH and pH Stability of Mutant Lipases

The optimum pH of the mutant lipases was determined by preparing the olive oil emulsion for the assay with 50 mM buffer systems of different pH (ratio 1:1); namely sodium acetate pH 4.0 to 6.0, sodium phosphate pH 6.0 to 8.0, Tris-HCl pH 8.0 to 9.0, glycine-NaOH at pH 9.0 to 11.0 and disodium phosphate pH 11.0 to 12.0. The assay temperature for the activity assay was done at their respective optimum temperature.

For pH stability, the mutant lipases were treated with pre-incubation at 60 °C in 50 mM buffer systems of different pH; namely sodium acetate pH 4.0 to 6.0, sodium phosphate pH 6.0 to 8.0, Tris-HCl pH 8.0 to 9.0, glycine-NaOH at pH 9.0 to 11.0 and disodium phosphate pH 11.0 to 12.0. The assay temperature for the residual activity assay was done at their respective optimum temperature.

## 5. Conclusions

The rational design strategy of employing stability prediction for substitution at the N-terminal resulted in improved stability and change in catalytic activity of wt-L2. A8E was found to be the most stable based on the MD simulation analysis, followed by A8P, wt-L2 and the least was A8V. However, the experimental temperature characterisation showed that the mutant lipases were enhanced differently. A8E had the highest T_m_ but its activity was significantly limited compared to A8V. Despite being the least stable in the MD simulation, the catalytic efficiency of A8V was the greatest. Whereas the stability of A8P was second to A8E and its catalytic efficiency was only slightly better than A8E. Therefore, the stability of structure enables the lipase to withstand high temperatures, but only to a certain extent before it comes at the cost of the catalytic activity.

## Figures and Tables

**Figure 1 molecules-25-03433-f001:**
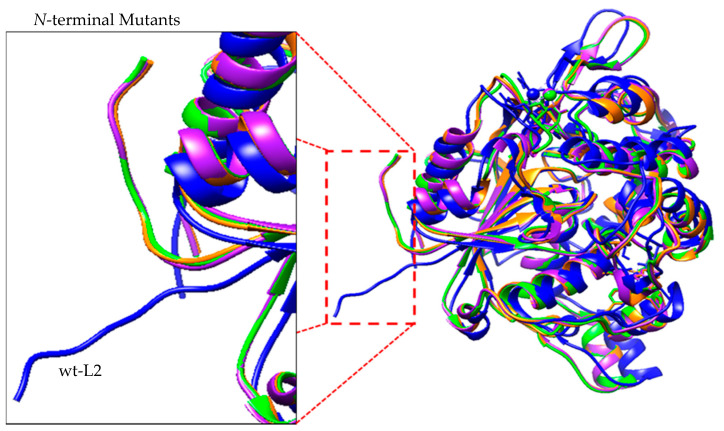
Superposition of global structure of wt-L2 (blue) with A8V (orange), A8P (purple) and A8E (green). Shown in the foreground is the anchoring of the *N*-terminal tail towards the global structure of the lipases. The anchoring is due to the formation of molecular interactions upon substitution at position 8.

**Figure 2 molecules-25-03433-f002:**
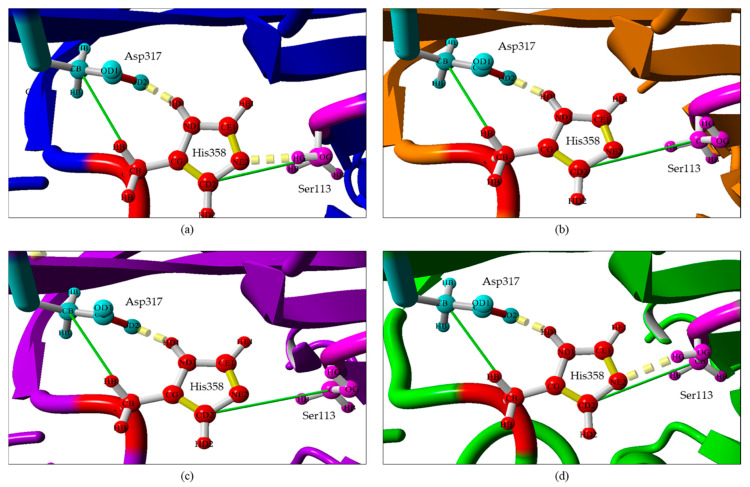
The hydrogen bonds (yellow dashes) and hydrophobic interactions (green line) in the catalytic triad of wt-L2 and mutant lipases. (**a**) wt-L2, (**b**) A8V, (**c**) A8P and (**d**) A8E.

**Figure 3 molecules-25-03433-f003:**
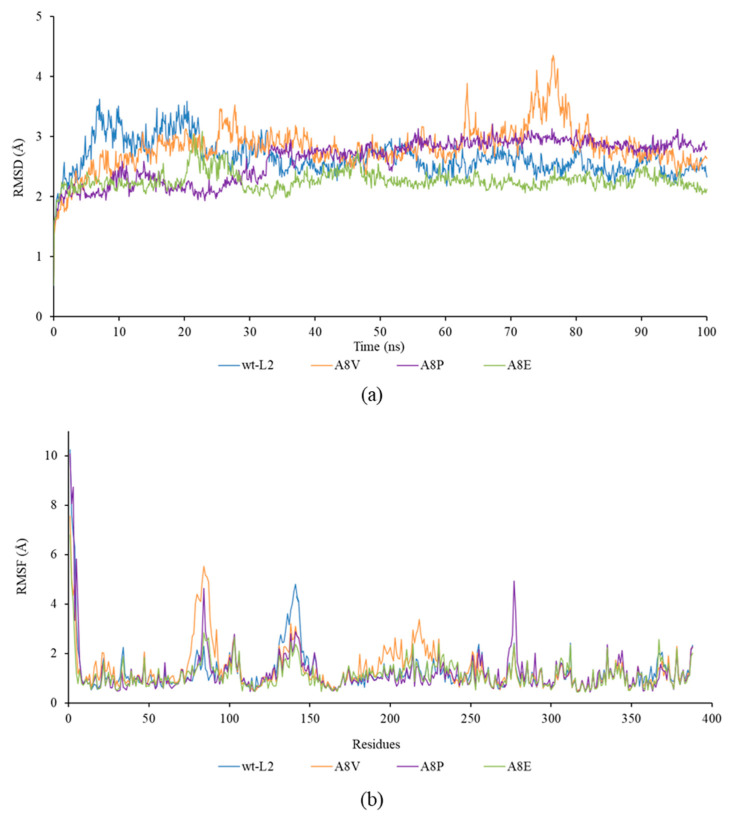
Molecular Dynamics (MD) simulation analysis of wt-L2 and mutant lipases for 100 ns using YASARA. (**a**) The root mean square deviation (RMSD) values. (**b**) The root mean square fluctuation (RMSF).

**Figure 4 molecules-25-03433-f004:**
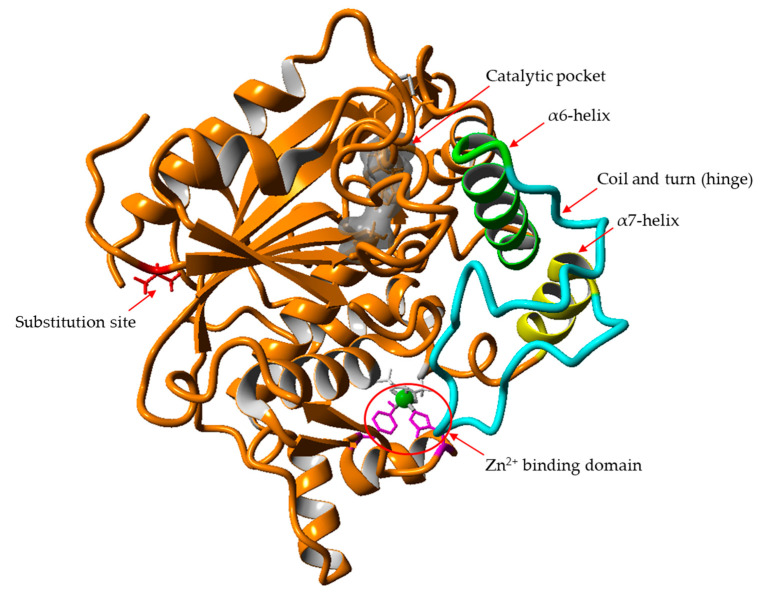
The relative position of the affected local structures with the substitution site in A8V.The flexibility of one of the lid helices α7 (yellow) and the coil and turn hinge (blue) increased upon substitution with valine at the critical point. The affected structures and the α6-helix lid (green) are involved in the interfacial activation mechanism. The affected Zn^2+^ binding domain is shown in magenta. Also shown is the catalytic pocket (grey) and the substitution site (red).

**Figure 5 molecules-25-03433-f005:**
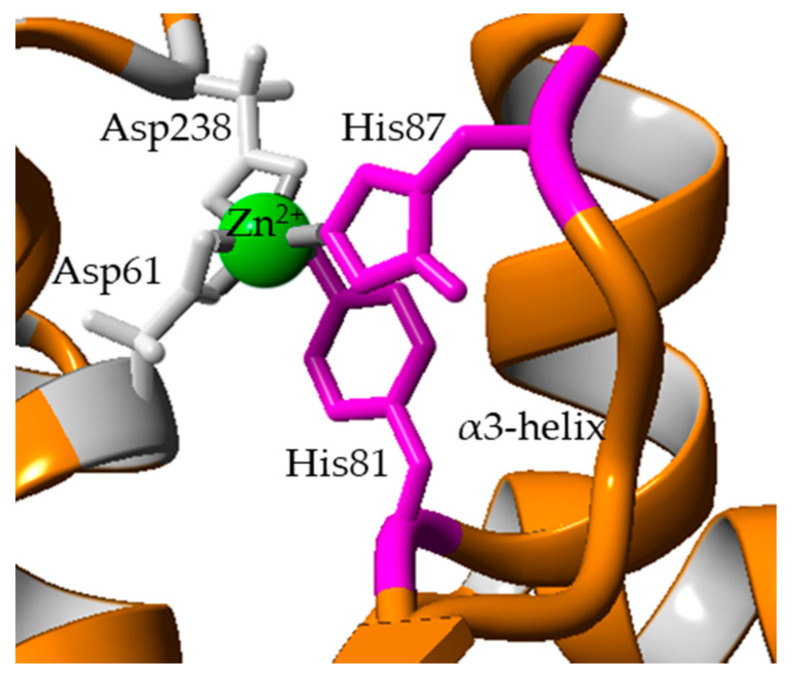
The Zn^2+^ binding domain by residues His81 and His87 (magenta) of the α3-helix. Shown in grey are Asp61 and Asp238, the other 2 residues in Zn^2+^ binding.

**Figure 6 molecules-25-03433-f006:**
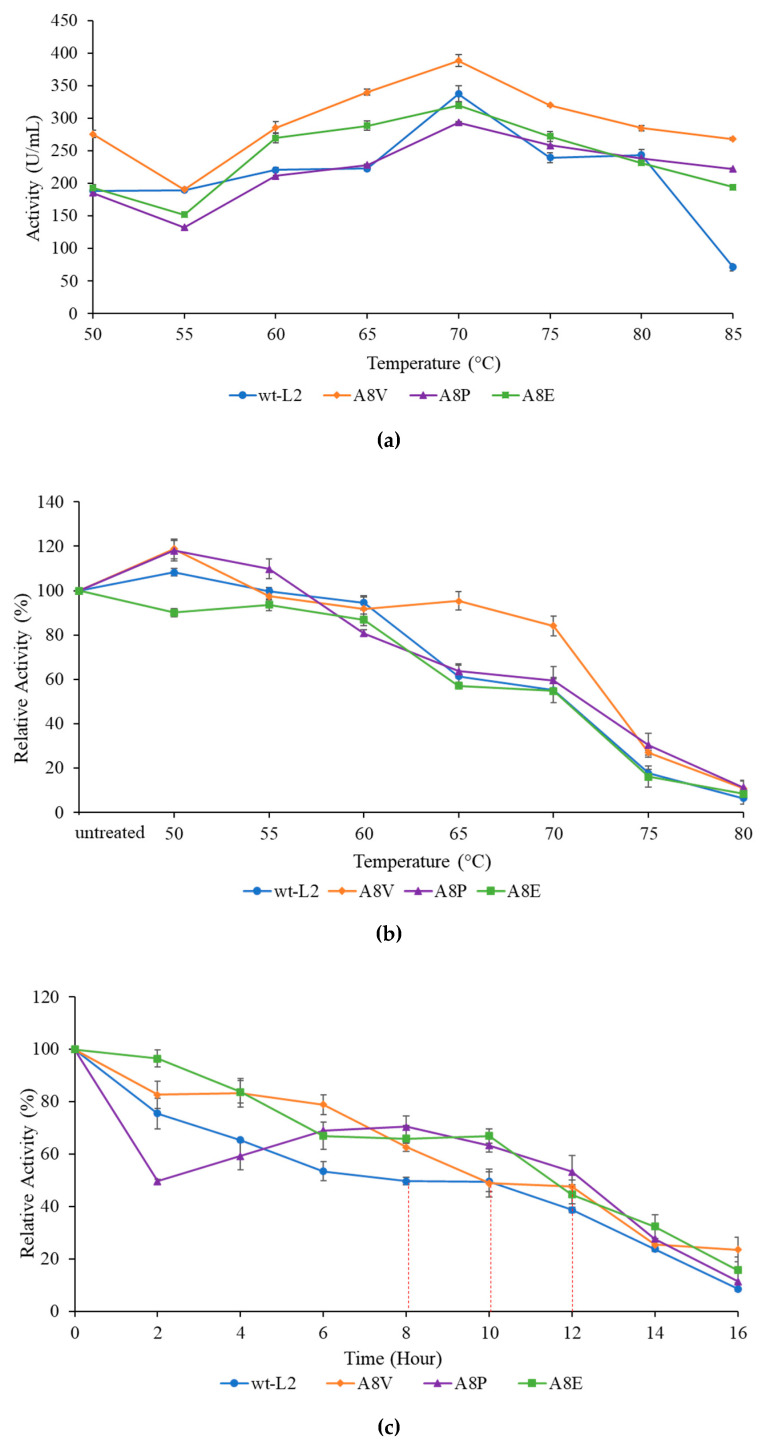
The activity profile of wt-L2 and mutants A8V, A8P and A8E. (**a**) The optimum temperature profile, (**b**) The thermostability profile and (**c**) The half-life profile. The relative activity was measured against untreated lipases and assumed to be 100%.

**Figure 7 molecules-25-03433-f007:**
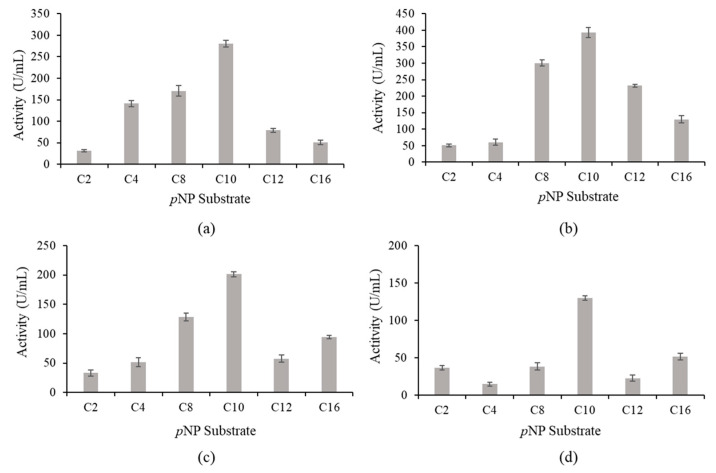
Substrate specificity of wt-L2 and mutant lipases was measured with *p*NP carbon acyl length between 2 to 16. (**a**) wt-L2, (**b**) A8V, (**c**) A8P and (**d**) A8E.

**Figure 8 molecules-25-03433-f008:**
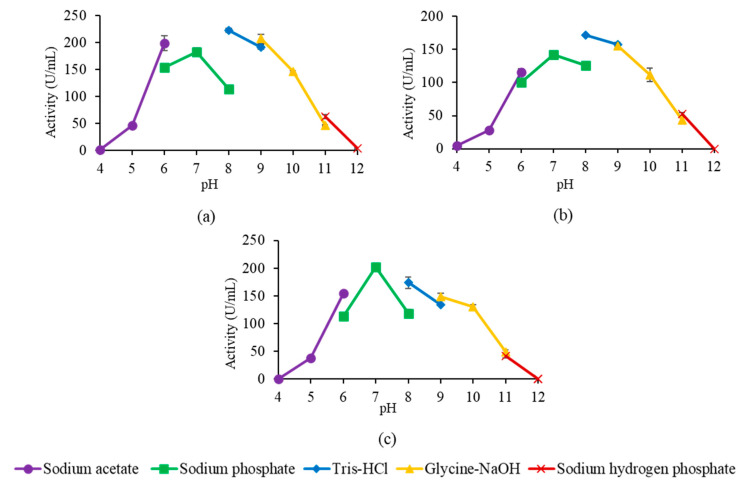
Effect of pH on the activity of mutant lipases. (**a**–**c**) pH profile of A8V, A8P and A8E, respectively.

**Figure 9 molecules-25-03433-f009:**
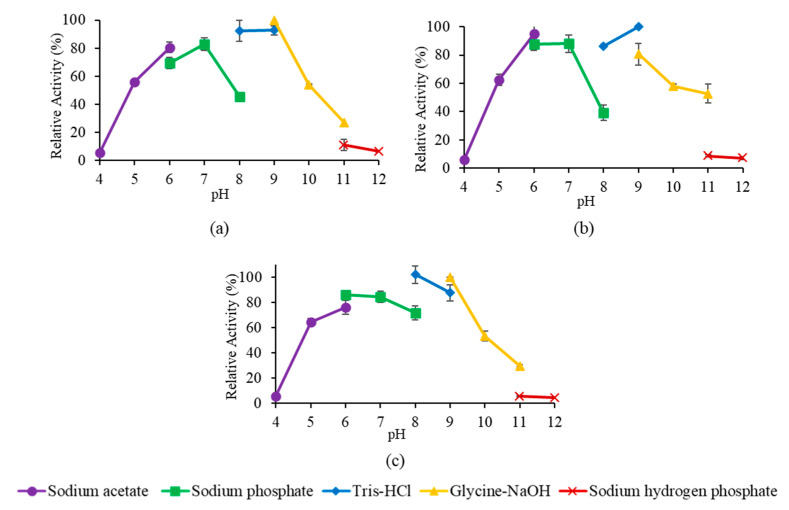
pH stability profile of mutant lipases. (**a**–**c**) stability profile of A8V, A8P and A8E, respectively.

**Table 1 molecules-25-03433-t001:** The number of hydrophobic, main chain-main chain and main chain–side chain interactions at the 11 potential critical points.

Position	No. of Potential Residues	No. of Interaction			Total Interactions
		Hydrophobic	Main chain–Main Chain	Main Chain–Side Chain	
Ala1	3	0	0	0	0
Ser2	14	0	0	0	0
Ala5	6	0	1	1	2
Asn6	9	0	1	1	2
Asp7	6	0	0	0	0
→Ala8	5	2	1	3	6
His14	7	0	0	2	2
Gly15	1	0	1	2	3
Thr17	6	0	1	1	2
Gly18	11	0	0	1	1
Gly20	11	0	0	1	1

**Table 2 molecules-25-03433-t002:** The stability change and Reliability Index (RI) score upon amino acid residue substitution at critical point Ala8.

Substituent	Stability Change	RI Score
→Val	Increase	5
Leu	Decrease	3
Ile	Increase	2
Met	Increase	1
Phe	Decrease	3
Trp	Decrease	4
Tyr	Decrease	4
Gly	Decrease	5
→Pro	Increase	6
Ser	Decrease	7
Thr	Decrease	7
Cys	Increase	1
His	Decrease	5
Arg	Decrease	1
Lys	Decrease	6
Gln	Decrease	6
→Glu	Increase	3
Asn	Decrease	4
Asp	Decrease	1

**Table 3 molecules-25-03433-t003:** The total number of hydrogen bonds and hydrophobic interactions at the *N*-terminal tail of wt-L2 mutant lipases.

Interaction	Total Interactions		
	A8V	A8P	A8E
Hydrogen Bonds	5	5	8
Hydrophobic Interaction	18	23	25

**Table 4 molecules-25-03433-t004:** The kinetic constants of wt-L2 and its mutants.

Lipase	Vmax (µmol/min/mL)	K*_M_* (mM)	*kcat* (min^−1^)	Catalytic Efficiency (s^−1^/mM)
wt-L2	105.26	1.08	10,526	162.43
A8V	81.30	0.52	8,130	206.57
A8P	76.33	1.34	7,633	94.93
A8E	71.91	4.38	7,194	27.23

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
