# Peer review of "Single Residue Substitution at N-Terminal Affects Temperature Stability and Activity of L2 Lipase"

_molecules, 2020, doi:10.3390/molecules25153433_

Round 1
Reviewer 1 Report
The work is very interesting, very well planned, very well structured and very well written.
Authors should review and change the names of the microorganisms in the text and place all in italics, as in line 48 (Bacillus sp).
Thus, I consider it suitable for publication in molecules.
Reviewer 2 Report
The authors apply a rational strategy to select positions at the N-terminus of wt-L2 in order to increase the stability of this interesting Lipase that belongs to the family I.5 of lipases. The mutations predicted indeed increased the stability of the enzyme. Then, the authors modeled the N-terminus of wt-L2 (missing in the crystallographic structure) and the different mutants in order compared them, infer and rationalize the potential structural changes. Although the simplicity of the strategy and the results are promising, I found some concerning flaws that need to be addressed before agreeing to publish this work. One of the main flaws, is the modeled N-terminus of the wt-L2 which does not seem to be consistent with all the structures previously solved from this family, and therefore the conclusions from such comparison lack reliability. The authors also showed a shockingly deficiency on the relevant structural information related to this family of lipases by falling to reference previous work, not making substantial and relevant comparisons to other structures in order to check the reliability of their models, and by overlooking pertinent information about the Lid and its interfacial mechanism that could help to convey their own results.
Major concerns:
- Bibliography is quite short, especially in the introduction, but across the manuscript.
The Authors should take the opportunity to be more extensive with their citations. They are doing a disservice to their own work by being so sparing with the references. There is wealth of knowledge that needs to be given credit while using it to educate the readers and convey their own story.
- When talking about the simplicity of your strategy, give a good contrast by citing/describing several other complicated strategies to increase stability and activity such as disulfide engineering (include the following references: Han et al.; 2009, Lu et al., 2012, Le et al., 2012; Godoy et al., 2019).
- Briefly mention and give more references to other mutation strategies on lipases.
- In the following sentence no reference was given: “The structure of L2 lipase was solved using X-ray crystallography and showed a 50 globular α/β hydrolase fold.” Please do
- Although this Lipase is from the I.5 family and other structures have been solved, which would help the readers to understand how they generally behave, but no references are given, nor they are even mentioned once.
- The authors wrote: “Despite many studies performed to understand the relationship between the characteristics of a thermophilic lipase and the impacts on stability and activity, the contribution of the terminal region remained mostly unexplored” and not even one of such studies was referenced (see references above which should all be included).
- The open-closed mechanisms that could have implications in the way their mutations behave, and for which the I.5 family has unique features, is not discussed. Not discussing is perhaps fair, but not even mention it along the whole manuscript is again a disservice to their own work. It should be at least mentioned to help the reader understand the rich mechanistic behavior of this family of lipases and the proper references given (Carrasco-Lopez et al., 2009).
This was the first red flag and illustrates how the authors later failed to check obvious flaws in their story. Unfortunately, if researchers are -willingly or not- prone to omit good part of the available body of references, then they are also likely to oversight rational considerations from their own results that could be easily spotted by considering what previous work have to teach.
- The RMSD values of the superimpositions of the wt-L2 vs the single modeled mutants, are extremely high (all over 2A). For a RMSD, a value over 2A generally means that there is a considerable structural change whether it is focused to a particular region or extended along the structure. As an example, the RMSD of the superimposition of wt-L2 and T1 Lipase (which the authors used to model the N-terminus of wt-L2) is way less than 1A. Even more interesting is that its superimposition with BTL2, which is a lipase from the same family I.5, but crystallized in the open conformation, is a little higher but also less than 1A. So, how the same structures with a single mutation can be over 2A? I can see that in the superimposition shown in Fig 1, that there is a considerable mismatch along the full-length of the structure for some mutants. How is this possible? We are talking about single mutants in the N-terminus. Can the authors really trust such differences? Perhaps there is something fundamentally wrong with either the superimpositions or the mutant models.
- Also, the extended N-terminus of the wt-L2, which was also modeled, does not make any structural sense. Is not making any contacts, how can it be stabilized in such position? The fact that the N-terminus (first 7 residues) it is not seen in the crystal structure is already an indication of its flexibility, so how the authors can assure that a reliable structure of the wt-L2 N-terminus is indeed stable in such position with no interactions? For anyone with any experience in protein structures should be extremely hard to believe this. As this was modeled using another structure did the authors consider than perhaps the N-terminus of the structure used as template has this extended conformation due to contacts within the crystal?
- To continue with this line of thought, something that is even more concerning, is that both the structures of T1 lipase and BTL2, which are not truncated at the N-terminus (well, only missing the first residue), have their N-terminus is a conformation that is remarkably similar to that of the mutants shown here but not at all to the modeled wt-L2. How is that possible if the authors used T1 lipase as template to model the N-terminus of their wt-L2? Either the authors chose the wt-L2 model that had the biggest difference with their modeled mutants, or the authors failed to perform a simple and obviously needed check comparing the final conformation of the N-term their modeled wt-L2 to that of the structure they used as template and other available structures such as BTL2. Moreover, position 8 in BTL2 is an aspartic acid while in wt-L2 and T1 is Ala. Nevertheless, both N-terminus of BTL2 and T1 as just mentioned, have exactly the same conformation (docked against their core as the mutants shown in this study). So how comes the conformation of wt-L2 is so different, have the author a hypothesis to explain this fact? Is the rest of the N-term so different among them to explain it?
- As the sequence and structure of lipases from the I.5 family are so conserved, the authors should make a comparison of wt-L2 with all the I.5 family structures available (not that many), to contrast and check how reliable the wt-L2 N-term model is at the N-terminus. Also, they should make a comparison of the mutants with the other structures from the I.5 family to check whether the differences they claim are consistent.
- The authors should low down their claims of the atomic differences within the active sites. They are extracted from a comparison of models in which the only difference is one residue far away from the catalytic pocket, and especially if they claim such subtle differences with sub-angstrom resolution. The authors should make extremely clear that the observations are within the context of comparing Models.
- What the authors describe as α-12-Helix is actually helix α-7, which together with α-6-Helix, and the “coil and turn” (anchored by the Zn+2) are all part of the Lid in the I.5 family of lipases (primary lid, α-6, and secondary lid, α-7). This has been reported and very well described before and the authors would know it if they have checked the appropriate literature (Carrasco-Lopez et al., 2009), when talking about the interfacial activation. Referencing reviews, as the authors do in this particular case, is always valid, but getting all the information from third sources, instead of the original ones is, the least, risky and as in this case an easy way to overlook important information. Furthermore, although a lot is said about the flexibility of the “coil and turn” from the simulations of the mutants, again the uniqueness of the Lid movement from the I.5 family of lipases is never mentioned, therefore disconnecting their results from the enormously relevant particularities if this phenomenon and the potential correlations with their results.
- Please double check language, I have found a consider number of grammatical errors and in general some parts are not quite clear and hard to follow.
Minor concerns
- Please be careful when you define functional and not functional regions as something generalizable to all proteins. Why the N-term is considered a non-functional region in general? The authors most probably know that in many examples the N-terminus can be part of the active sites, a hinge for structural changes, an anchor to the membrane or the key element in protein-protein interactions. So, please be clear and reformulate the way you define functional/non-functional regions or make sure you are being specific to this particular Lipase or Lipases from the same I.5 family and not all proteins in general which is what is implicit in the authors statements, particularly in the Abstract but also in the introduction.
- The authors stated: “The N-terminal region of L2 lipase was defined as amino acid residues from Ala1 to Gly20” Please be clear, who defined this and based on what? It is a very different statement if the authors say something like: In this study, the N-terminal was defined as…
- The subtitle “Structure Comparison of wt-L2 and mutant lipases” is misleading because none of the mutant structures were solved but they are models used to compare with the crystallographic structure of wt-L2 which N-term was modeled as well. Then it should be changed to “… and mutant lipase models”.
- Figure 4 does not have a title. Please add a title to the figure.
- In the results no explanation whatsoever given on how the models for the mutants were made. Please briefly explain that before comparing with the wt-L2, that would help to avoid misleading interpretations and for any details the readers can check the Methods…
- Numbers and labels in Figure 6 are way too small. Please edit accordingly.
Reviewer 3 Report
The comparison between computational predictions and experimental validation is always important, both for feedback and also to improve future research. In particular, the interest in lipases is scarce compared to proteases or other enzymes, from basic and biotechnological approaches.
The selection of the mutants is adequate and the rationale of studying the N terminus is interesting.
The prediction of hydrogen bonds from the simulations would be questionable, but if is fine if it correlates with structural stability. Figure 2 for showing hydrogen bonds is too simplistic, see many structural papers to see how a hydrogen-bonding pattern is presented. Use Pymol or some other protein rendering program, figure 2 is extremely simplistic, what structural elements are in contact
for example, in figure 2 (b), how the hydrogen bond between S113 and H358 is lost, without considering bulk water? The picture they present is too simplistic. If in a protein structure there is uncertainty sometimes positioning hydrogen bonds, in a simulation much more.
Keep one of the most important MD plots and send other plots to the supplemental data.
Please correlate the substrate specificity with the active site cavity. There is a structural reason why C10 is preferred over C16 or C2 or C4?
The title of table 4 does not match the kinetic data. Please elaborate on the loss of activity for the A8E mutant and the balance between loss of affinity or kinetics. What is the most important aspect?
The ellipticity vs. temperature data can be used in expressing thermodynamical stability. There are several papers to check for this. Besides specific activity, the mutations stabilize the lipase structure?
https://pubmed.ncbi.nlm.nih.gov/24391996/?from_term=lipase+mutant+circular+dichroism&from_pos=10
Please include an SDS-PAGE gel for expression and purification as supplementary data. For the native and the mutant proteins too.
Please include as supplementary data the raw CD files.
Figures in the appendix should be improved, the legends are hard to read.
Round 2
Reviewer 2 Report
The authors have addressed almost all my concerns. Although I am still not convinced by the modeled conformation of the N-term in wt-L2, I do understand and agree with the authors that it is a checked model and also not the major point of the manuscript, which is instead the remarkable improvement of thermostability and half-life by single mutations at the N-term. There is no reason to delay the publication due to this concern. Nevertheless, I do have a couple of minor revisions that MUST be taken into account before publishing and I trust that the authors will address them in full and the editors will make sure this is the case.
Minor revisions to improve it further:
- Authors should revise the following sentence in the abstract in lines 17-18:"However, few explored the role of the non-functional region" for something like "However, few have explored the role of other regions which, in principle, have no evident functionality". This will help to avoid any posible misinterpretations.
- The designations of the secondary structure and composition of the lid found in Figure 4 and lines 166-169, should be homogeneous and based on the latest information (which is actually not that recent) as a member of the I.5 family. That is how science works and advance. Reference 28 is at least 6 years previous to the references that actually described how the Lid works and its secondary structure in the open conformation (29 and 33), so there was no way they could have known that it was a double lid at that time. Unfortunately, this also means that the authors of references 24 and 34 (mostly the same; also includes authors of the study under review here) which were published after 29, failed to check and reference at the time (as they did here as well) the latest advances for the I.5 family to correctly rename the Lid, an error that should not be carried any further. The I.5 family is highly conserved, therefore the double Lid is also conserved and should be described as in reference 29 because, with great certainty, it is the exact same for every single member of the family. This is the most basic way science advances, by embracing the new data when is evident that is true, so please, do not continue to use an obsolete way to described the I.5 Lid just for the sake of referencing yourselves. This will ultimately confuse the readers. The authors made an unfortunate omission on those publications (24 and 34) and also in here, I understand it happens, but now that the authors know, it SHOULD BE CORRECTED IN BOTH FIGURE 4 AND THE MANUSCRIPT, otherwise is not an honest mistake anymore.
